# Adhesion Effect on the Hyperfine Frequency Shift of an Alkali Metal Vapor Cell with Paraffin Coating Using Peak-Force Tapping AFM

**Jiuyan Wei** [1,2]**, Zongmin Ma** [1,2,*]**, Huanfei Wen** [1,2]**, Hao Guo** [1,2]**, Jun Tang** [1,2]**, Jun Liu** [1,2,*]**, Yanjun Li** [3] **and Yasuhiro Sugawara** [3]

[1] National Key Laboratory for Electronic Measurement Technology, North University of China, Taiyuan 030051, China; davidjiuyanwei@163.com (J.W.); hfwende@163.com (H.W.); guohaonuc@163.com (H.G.); tangjun@nuc.edu.cn (J.T.)

[2] Key Laboratory of Instrumentation Science & Dynamic Measurement, North University of China, Ministry of Education, Taiyuan 030051, China

[3] Department of Applied Physics, Osaka University, 2-1 Yamada-Oka, Suita, Osaka 565-0871, Japan; liyanjun@nuc.edu.cn (Y.L.); sugawara@nuc.edu.cn (Y.S.)

[*] Correspondence: mzmncit@163.com (Z.M.); liuj@nuc.edu.cn (J.L.); Tel.: +86-0351-3922399 (Z.M. & J.L.)

**Abstract:** We have investigated the adhesion effect on the hyperfine frequency shift of an alkali metal vapor cell with paraffin coating using the peak-force tapping AFM (atomic force microscopy) technique by developing a uniform and high-quality paraffin coating method. We observed a relatively uniform temperature field on the substrate can be obtained theoretically and experimentally with the closed-type previse temperature-controlled evaporation method. The roughness and adhesion of the coating surface as low as 0.8 nm and 20 pN were successfully obtained, respectively. Furthermore, the adhesion information dependence of the topography was investigated from the force spectroscopy, which indicates that the adhesion force jumped on the edge of the particles and stepped but remained constant above the particles and steps regardless of their height for paraffin coating. Finally, we can evaluate the relaxation and the hyperfine frequency shift of an alkali metal vapor cell through accurately calculating the surface adsorption energy of the paraffin coating from peak-force tapping information. This finding is crucial for improving the sensitivity of the atomic sensors through directly analyzing the adhesion effect of the paraffin coating films instead of measuring the relaxation times.

**Keywords:** peak-force tapping; alkali metal vapor cell; adhesion effect; atomic force microscopy

## 1. Introduction

Coated alkali vapor cells with paraffin coating have been extensively investigated because of their long spin-polarization lifetime and high detection sensitivity for application in a wide range of atomic sensors, such as atomic magnetometers, atomic clocks, and magneto-optical traps [1–4]. The paraffin coating can extend dwell times of spin-polarized Rb atoms to 1.8 μs compared to that of an uncoated cell [5], which can support approximately 10,000 polarization-preserving collisions with an alkali atom in cells [6–8]. Therefore, the coating of alkali metal vapor cells and its mechanism on alkali metal atoms have attracted extensive attention [8–10].

Conventional paraffin coatings can be obtained by casting and cold-finger methods, which form paraffin crystals with three stages: nucleation, growth, and aggregation or coalescence of the paraffin crystals [11,12]. A chemical vapor deposition system can also be used to deposit amorphous carbon-based films in methane ($CH_4$) gas [13]. However, it is reported that the paraffin coatings

produced using the above methods always result in uneven morphology, and their relaxation times of the alkali metal atoms are relatively short.

By contrast, the coating performance is mainly measured by the relaxation time of the alkali metal vapor cell through optical quantum detection [14–17]. Seltzer successfully developed an alkali test cell to test the antirelaxation properties of surface coatings and found that the antirelaxation quality changes as a function of temperature [7]. Bandi et al. successfully measured the $T_1$ and $T_2$ relaxation times to characterize the wall coating and found that a polarized Rb atom undergoes approximately 2255 wall collisions (at 300 K) before losing its polarized state [18].

Recently, some researchers began to investigate the coating performance beyond relaxation-time-combined surface analyses, such as morphology, infrared spectroscopy, and NEXAFS spectra [19–22]. Rampulla et al. investigated the morphology and composition of coated Rb vapor-exposed films and found that an alkali island formed on the surface could affect the antirelaxation behavior of organic coatings [1]. Seltzer et al. studied the light-induced atomic desorption yields combined with a variety of surface characterization methods and found that measuring and determining the crystallinity of the coating material is unnecessary [21]. However, there are still two points that researchers seldom concentrate on, which are especially critical for the sensitivity of the atomic sensors. One is the method of producing alkali vapor cells with paraffin coating, whose performance cannot be predictable during the fabrication process. The second point is the adhesion effect, occurring during the energy transfer in the process of surface adsorption [11], which seriously affects the hyperfine frequency shift closely related to the relaxation time for the alkali-metal atoms that interact with the paraffin coating.

In this research, we investigated the energy transfer of the substrate during the evaporation process by comparing three preparation methods and found the method of preparing a uniform paraffin coating. Furthermore, with the peak force tapping AFM (atomic force microscopy) technique, the relationship between the adhesion effect of the paraffin coating and the hyperfine frequency shift of the alkali metal vapor cell was studied. From the result, it can be seen that the surface of the paraffin coating prepared by the closed-type precise temperature-controlled evaporation method with uniform temperature field was the flattest and smoothest; the roughness was 0.214 nm, and the adhesion was 20 pN. Moreover, by force spectrum analysis, it is found that the adhesion jumps occur at the edge of the particles and the steps, and when the height fluctuation of the coating is less than 1 nm, the viscosity of the adhesion is lower than 6 pN. Compared with the silicon substrate, it was found that the paraffin coating can significantly reduce the energy loss caused by the adhesion. Finally, we calculated the surface adsorption energy of the paraffin coating by analyzing the energy transfer of the force curve measurement, which provides a new idea for estimating the relaxation and the hyperfine frequency shift of alkali metal vapor cells.

## 2. Experimental Details

Crystalline silicon (Si samples) supplied by Nano Material was used as substrates for paraffin evaporation. The surface of the substrate, with diameter dimensions of Ø 50.8 × 0.5 mm, was washed with isopropyl alcohol in an ultrasonic bath and then rinsed by deionized water [11]. The substrate's mechanical properties, given in Table 1, were supported by SCI-TECH innovation.

**Table 1.** Mechanical properties of silicon samples.

| TIR | TTV | Bow/Warp | $R_a$ |
|---|---|---|---|
| <3 μm | <10 μm | <40 μm | <0.5 nm |

Here, three methods to produce paraffin coating—namely, open-type evaporation, the closed-type rapid cooling evaporation, and closed-type precise temperature-controlled evaporation—are proposed. Closed-type precise temperature-controlled evaporation and closed-type rapid cooling evaporation are

collectively called closed-type evaporation. The paraffin used in this work is a mixture of long-chain hydrocarbons with chain lengths ranging from $C_{19}$ to $C_{49}$ with linear, ramified, even and odd chains [11].

The paraffin with the weight of 4 g was stored in a paraffin container and heated by the heating platform, which is under the paraffin container, as illustrated in Figure 1. Each evaporation device consists of a heating platform and a paraffin container with a silicon substrate fixed on it. Figure 2 illustrates the schematic of the closed-type precise temperature-controlled evaporation system. The evaporation system consists of a temperature probe, a voltage supply, a PC, an evaporation chamber, and a PI controller with integrated data acquisition. The temperature is measured by the PI controller using a probe placed on the sample inside the evaporation chamber. The proportional integral algorithm is used to calculate the drive current based on the deviation, and the upper and internal heaters are driven by the voltage supply to realize the internal high-precision temperature in the heat container.

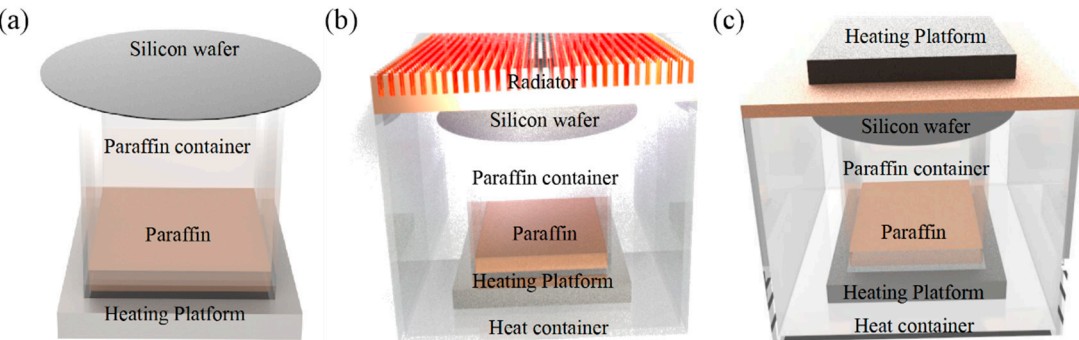

**Figure 1.** Three evaporation rigs of product paraffin coating: (**a**) open-type evaporation; (**b**) closed-type rapid cooling evaporation; and (**c**) closed-type precise temperature-controlled evaporation. Closed-type precise temperature-controlled evaporation and closed-type rapid cooling evaporation are collectively called closed-type evaporation.

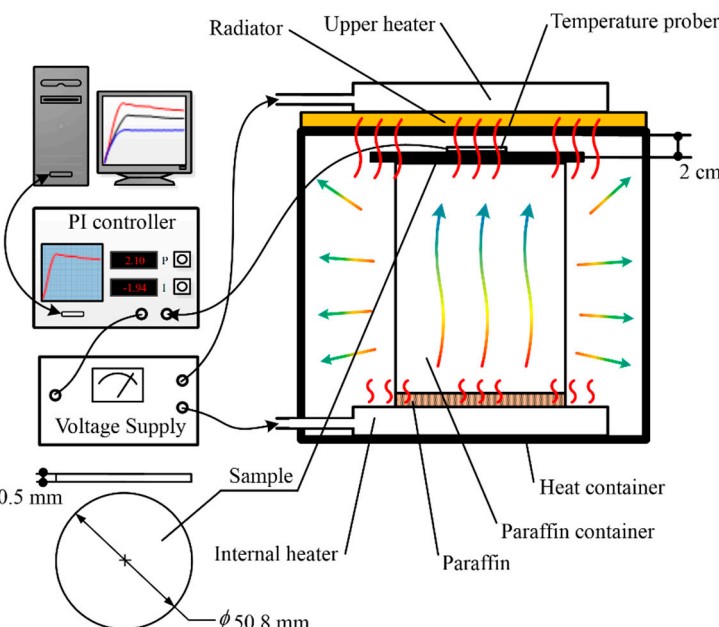

**Figure 2.** Schematic of the closed-type precise temperature-controlled evaporation system.

After preparation, the specimens were submitted to surface characterization using an ICON (Bruker, Germany) AFM. The AFM operating in the peak force tapping (PFT) mode was applied to investigate the topography and adhesion of paraffin coating on Si substrates [22–27]. To ensure

the accuracy of the experimental results, the cantilever spring constant was calibrated using a pure crystalline silicon sample and a calibration algorithm prior to peak force measurement [28–30].

Figure 3 illustrates the force curve and mechanical properties that can be obtained from it in the peak force tapping mode. The below red line indicates the measured force in the process of approaching the sample, and the corresponding black line is the measured force in the process of withdrawing from the sample. When the probe approaches the sample from a distance, the probe is subjected to the attractive force (mainly van der Waals force and electrostatic force) as a negative force. The repulsive and attractive force reach equilibrium at point A. Then the force between the probe and the sample reaches the peak at point B, which maintains constant in PFT mode through system feedback [29]. In the withdraw process, the force shows a difference between point C and point B, which were called pull-out points. Additionally, the force spectrum information of the surface can be obtained by measuring the force spectrum for the surface, and the adhesion, dissipation, deformation, and other information of the sample can be obtained by a fitting analysis.

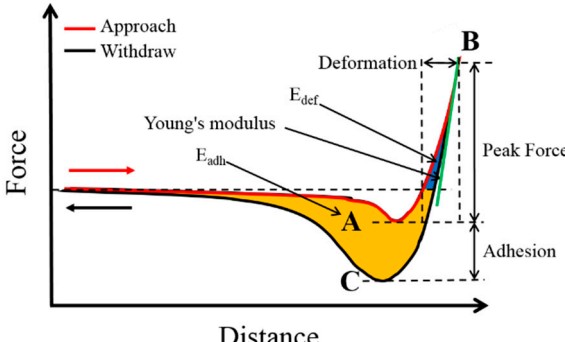

**Figure 3.** Force curve and mechanical property that can be obtained from them when operated in PFT mode. The red line is the process of approaching. After arriving at point B from point A, the black line indicated the withdrawn process from point B to point C.

## 3. Results and Discussion

First, we performed peak force tapping AFM measurements of the paraffin coating produced by open-type evaporation, closed-type rapid cooling evaporation, and closed-type precise temperature-controlled evaporation, which illustrated in Figure 4a–h. Herein, Figure 4a–c is the topography and Figure 4d–f is the corresponding adhesion information, which measured at the same time. Figure 4g,h is the cross-section of the white line in Figures 4a–c and 4d–f.

The topography of the paraffin coating produced by three methods of deposition, as illustrated in Figure 4a–c,g, the various paraffin steps combined with different shapes of paraffin particles can be observed. In Figure 4a, paraffin particles are scaly; it is due to the difference in temperature field during preparation formed surface with a step of approximately 115 to 200 nm and roughness of 78.9 nm. By contrast, in Figure 4b, the surface of the coating prepared by closed-type rapid cooling evaporation is relatively uniform with a particle height of 12 nm and roughness of 0.871 nm. Besides, many protrusions of 3 nm and 5 nm were observed on the flat paraffin surface, which was affected by the temperature and fluid fields during the cooling process [6].

On the other hand, in Figure 4c, the crack-like traces are manifested during the slow cooling process, and a uniform paraffin coating surface with a height difference of approximately 0.8 nm and roughness of 0.214 nm was formed. The formation of uniform surface is due to the temperature of the sample produced by closed-type precise temperature-controlled evaporation is stable and appropriate, the paraffin coating on the surface diffuses secondly, which grow governed by the mass transport [3]. Comparing these three coatings surface, in Figure 4g, the step height of coating produced by the open-type evaporation is 14-times higher than that provided by closed-type rapid cooling evaporation

with 12 nm, and 143-times higher than that provided by closed-type precise temperature-controlled evaporation with 0.8 nm, which is depicted in the inset.

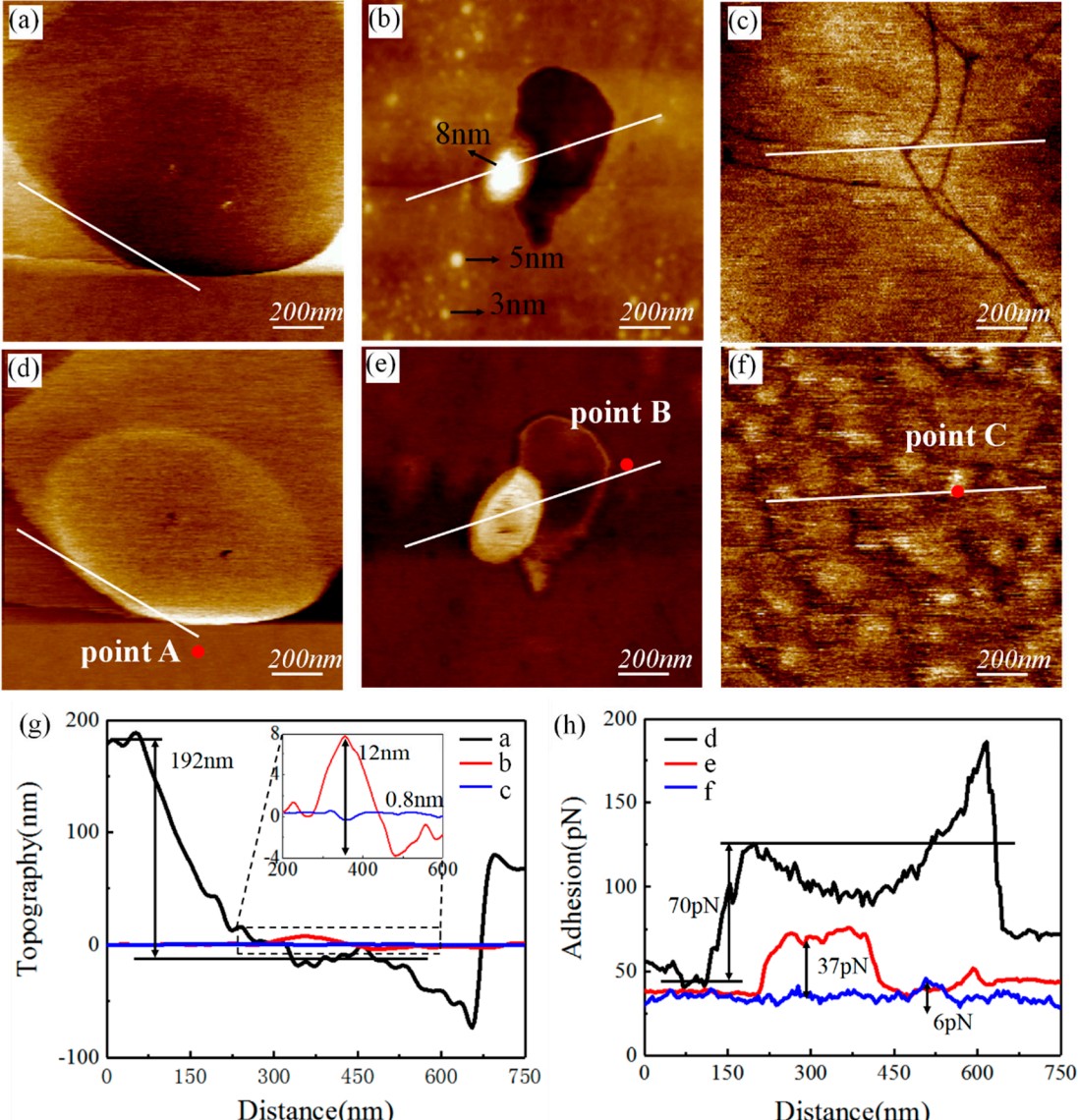

**Figure 4.** Simultaneous measurement of topography and adhesion in the peak force tapping mode: (**a**–**c**) topography of paraffin coatings produced by open-type evaporation, closed-type rapid cooling evaporation, and closed-type precise temperature-controlled evaporation (white lines depict the cross section of extracted profiles); (**d**–**f**) adhesion of paraffin coatings produced by open-type evaporation, closed-type rapid cooling evaporation, and closed-type precise temperature-controlled evaporation (white lines depict the cross section of extracted profiles); (**g**) cross section of the topography corresponding to (**a**–**c**); and (**h**) cross section of the adhesion corresponding to (**d**–**f**). Each paraffin coating was evaporated for 75 min.

In Figure 4d–f,h, the adhesion information corresponding to Figure 4a–c,g can be seen, the point A, B, and C are the force spectrum measurement points will be discussed later. In Figure 4d, the adhesion is stepped and fluctuates greatly, it shows the same shape of paraffin step as the topography, and the minimum viscous force is approximately 50 pN. The adhesion of bulging paraffin particles is 70 pN, corresponding Figure 4a shows the step height with 192 nm, suggesting that the changing morphology will result in greater viscous forces. This behavior is also observed in Figure 4e, around the 8 nm

paraffin bulge, a sudden change with 38 pN in the adhesion. Particularly, adhesion of the edge of the dark region, where the adhesion in the center is weak and uniform as 70 pN, shows a jump of 10 pN, while the insert in Figure 4g shows the height difference of the center defect as 4 nm. However, in Figure 4f, the adhesion is different from the topography as adhesion shows a granular distribution and fluctuation of 6 pN in Figure 4h while the topography is relatively flat with fluctuation of 0.8 nm in Figure 4c, suggesting that adhesion can see the defects that the topography cannot. In addition, in Figure 4h, it can be found that the adhesion of the paraffin coating produced by the open-type evaporation is 1.9-times higher than that provided by closed-type rapid cooling evaporation and 11.7-times higher than that provided by closed-type precise temperature-controlled evaporation. While the minimum adhesion of the coatings produced by three methods are similar as 30–50 pN.

The formation of paraffin coating is governed by the growth of paraffin particles affected by preparation temperature, which involved in the temperature field and fluid dynamics [13]. Therefore, we simulate the temperature of the three preparation methods to study the paraffin production process, as shown in Figure 5. A massive temperature difference from the comparison of these three models is observed, wherein the temperature in the central region of the silicon wafer produced by the open-type evaporation method (Figure 5a) is the most intense, about 120 °C. This is the main reason for the formation of the overlapping distribution. As the temperature of the silicon wafer is approximately 45 °C lower than the energy required for diffusion, there will be no displacement and phase transformation after the solid paraffin particles have been attached to the silicon wafer. With the paraffin particles adhering to the surface non-uniformly, a layer covered structure surface is generated, and the attached paraffin particles have large sizes, resulting in a 100 nm paraffin block, as illustrated in Figure 4a.

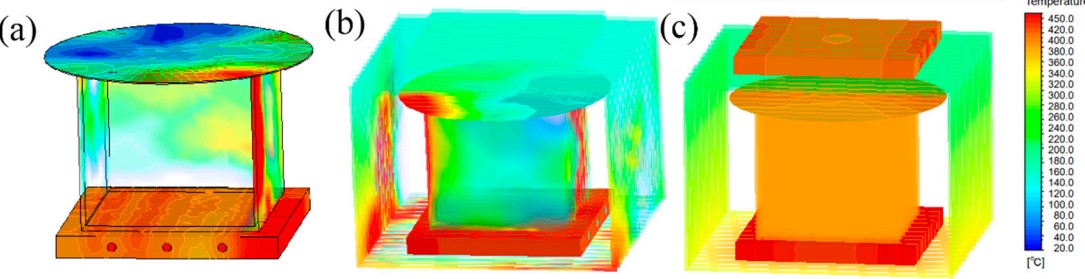

**Figure 5.** Distribution of temperature with three evaporation methods: (**a**–**c**) open-type evaporation, closed-type rapid cooling evaporation, and closed-type precise temperature-controlled evaporation. The temperature differences at the center of the sample are 120, 30, and 10 °C.

In contrast to open-type evaporation, closed-type evaporation achieves much more uniform temperature on the silicon wafer. In Figure 5b,c, the maximum temperature difference in the central region is 50 °C. As illustrated in Figure 5b, the temperature difference in the center of the sample is approximately 30 °C, which is due to the heat container and temperature controlled by the radiator above the silicon substrate and the bottom heater. However, during the cooling process, the radiator will lead to temperature fluctuation, which results in some surface defects. As illustrated in Figure 4b, paraffin particles with a size of 5 nm or 3 nm are distributed on the uniform paraffin coating. It is probably due to the temperature fluctuation during surface cooling, which results in the formation of some small paraffin particles. In addition, in Figure 5c, the closed-type precise temperature-controlled evaporation can keep the temperature fluctuation below 10 °C and maintain high temperature by the upper and lower heater so that it is conducive to the formation and merging growth of the surface paraffin island [13]. Although the high temperature is good for adherent paraffin particles to distribute by breaking the surface diffusion barrier, it requires a stable temperature and a slow cooling process.

Furthermore, the closed-type precise temperature-controlled evaporation with the best uniformity of temperature is taken as the research object. As illustrated in Figure 6, Figure 6a,b is the topography

and adhesion of the coating evaporated for 8 min, Figure 6c,d is the coating evaporated for 25 min. Figure 6e is the cross-section of the white line in topography Figure 6a,b, Figure 6f is the cross-section of the white line in adhesion Figure 6c,d.

With evaporation for 8 min, in Figure 6a, the independent paraffin particles with a size of approximately 23–45 nm, such as Particle I, are dispersed on the silicon substrate. Meanwhile, aggregated paraffin clusters (blue circle region) appear, due to the proper temperature of the substrate, which results in the migration and aggregation of paraffin particles by breaking the surface diffusion barrier. After evaporation for 25 min, in Figure 6b, the paraffin cluster continues to grow, which becomes broader and lower, as Particle II shows. Moreover, outside the cluster, the holes defects of petal shape can still be seen. It suggests that the paraffin grains form the nucleus and begin to grow by engulfing adjacent nuclei, which will form paraffin clusters and diffuse, then the grooves between paraffin blocks appear and finally forms a continuous and uniform coating [31].

In Figure 6c, the adhesion of the paraffin particles is approximately 26 pN, and on the contrary, the silicon substructure shows high adhesion as 80 pN. Therefore, it reveals that paraffin coating on the inner wall of the alkali-metal vapor cell can reduce the adhesion of the surface so that alkali atoms can spend less energy to overcome the surface attraction and increase the spin relaxation time. On the other hand, as paraffin cluster diffusing, the adhesion becomes flat with fluctuation of 17 pN, in Figure 6d. However, in Figure 6e,f, it can be observed that the paraffin particles with different heights (from 9 to 31 nm) have the same adhesion force, which is approximately 26 pN, while the peak of adhesion (green circle area) appears as approximately 150 pN at the junction of paraffin particles and the substrate. This result indicates that the adhesion of paraffin coatings with different thickness is the same, but the uneven coating will result in high adhesion.

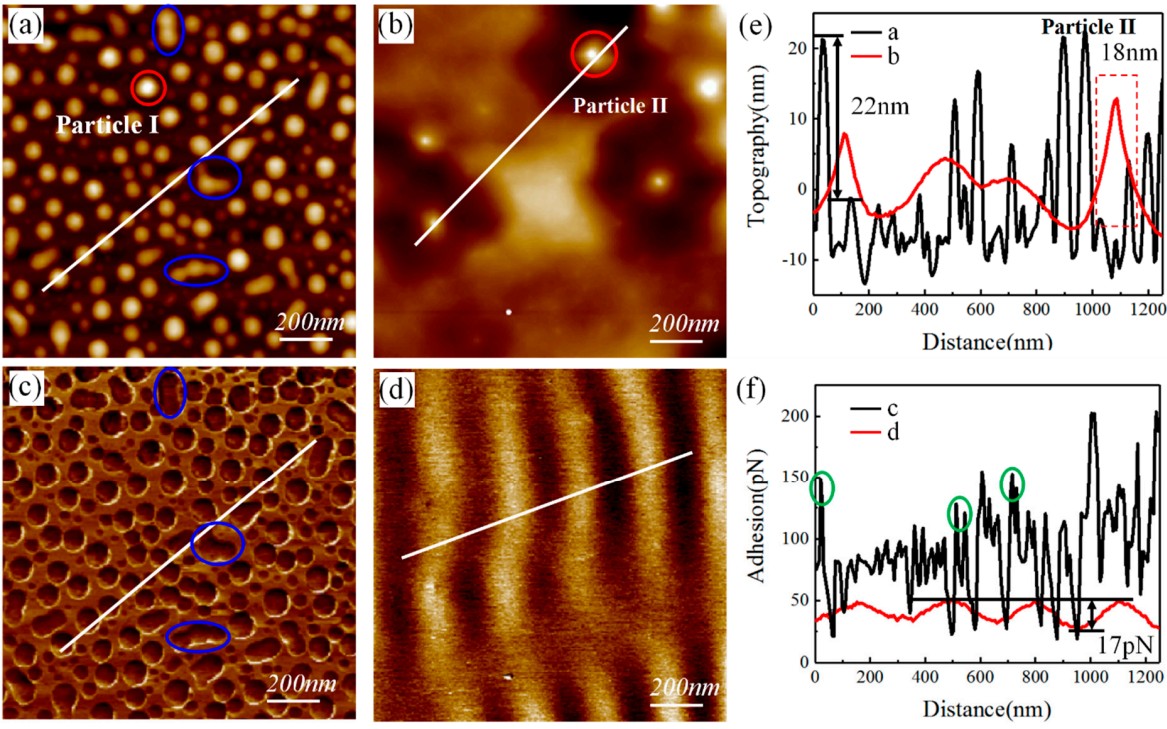

**Figure 6.** Paraffin coating produced by closed-type precise temperature-controlled evaporation for 8 min and 25 min: (**a**) topography with evaporation for 8 min; (**b**) topography with evaporation for 25 min; (**c**) adhesion corresponding to (**a**); (**d**) adhesion corresponding to (**b**); (**e**) cross section of the topography corresponding to (**a,b**); and (**f**) cross section of the adhesion corresponding to c and (**d**). The blue circled areas are gathered paraffin particles.

Paraffin coating can reduce the adhesion of the inner surface of the alkali-metal vapor cell, which can extend spin relaxation time and increase the number of collisions by reducing the energy loss during a collision. The collision between alkali atom and the coating is equivalent to the measurement of force curve, which can reveal the energy conversion by measuring the force with the outermost atom of a sharp tip in the process of approaching and withdrawing from the coating surface. Figure 7 shows the force curves we obtained on the three paraffin coatings from point A, B, and C in Figure 4 using peak force tapping technology. The black line is the approach process when the tip gets close to the sample, and the red line is the withdraw process after the tip reaches the setting peak force value. It can be observed that, in Figure 7c, the coating produced by closed-type precise temperature-controlled evaporation indicates the least adhesion with 26 pN compared with 81 pN in Figure 7a and 47 pN in Figure 7b. As the tip approaches the sample, the outermost atoms at the tip begin to be attracted by the sample at L1. Then the total attraction and repulsion reach the balance of 41 pN at L2. Moreover, the repulsion reaches the maximum at the total force of 73 pN. Subsequently, in the withdrawing process, the approach-to-withdrawal force curve shows a difference at L2 affected by the adhesion of the coating [23,26,28].

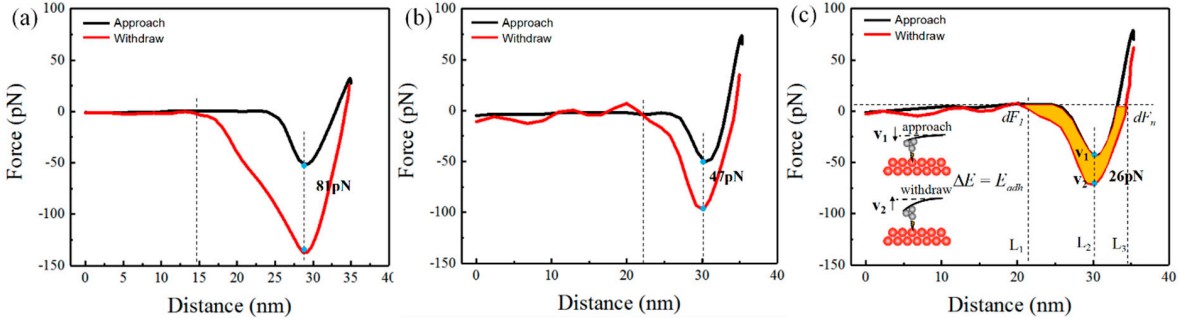

**Figure 7.** Force curve at three paraffin coatings produced by open-type evaporation, closed-type rapid cooling evaporation, and closed-type precise temperature-controlled evaporation, respectively: (**a**–**c**) is measured at points a–c in Figure 4.

In the measurement of force curve, the probe was driven by external energy ($E_{ext}$) with a sinusoidal function $z = z_0 + A\cos(\omega t + \phi)$ [28], and the excitation energy must be converted into either hydrodynamic damping in the medium ($E_{med}$) or energy dissipated in the sample ($E_{dis}$) [29]. Then, the dissipation on the sample can be calculated with Equation (1)

$$E_{dis} = E_{ext} - E_{med} = \frac{\pi k A}{Q}\left(A_0 \sin\phi - \frac{A\omega}{\omega_0}\right) \qquad (1)$$

where $E_{med}$ is modeled by a linear viscous damping law: $F_{med} = -b\dot{z}$. $Q$ is the quality factor of the cantilever; $A_0$ is the amplitude very far from the surface; and $Ø$ is the phase shift between the external excitation and the tip response. In the Figure 7c, we see that the energy loss caused by the adhesion in the interval $L_1$ to $L_3$ is defined as $E_{adh}$ [30,32], and the energy loss caused by an elastic collision can be calculated with Equation (2)

$$\Delta E = E_{adh} = E_{dis} - E_{def} = \sum_{i=1}^{n} dF_i \cdot dz \qquad (2)$$

where $E_{def}$ is the energy loss due to deformation caused by probe penetration into sample surface. By calculating with Equation (2), we find that the $\Delta E$ of three paraffin coatings can be obtained as 7.1725, 2.462, and 0.875 eV. The difference of adsorption energy is due to the different surfaces of the prepared coatings. This matches the order of wall adsorption energy of the paraffin coatings measured by Bouchiat and Brossel at 0.1 eV [33,34].

Therefore, we can conclude that using the peak force tapping technique, we can accurately calculate the adsorption energy of the coating, and the experimental results indicate that coatings prepared from the same paraffin with surface defects (bulges and steps) lead to the increase of the adhesion force and the increase of the surface adsorption energy. Moreover, the average adsorption time also increases as $\bar{t}_w = \tau_0 e^{\Delta E/kT}$, resulting in an increase in the ground-state hyperfine frequency shift as $\Delta v = \frac{\Delta E_{hf}}{h} \cdot \frac{\bar{t}_w}{\bar{t}_c}$ [35–38], which is directly related to the detection sensitivity of the atomic sensor.

## 4. Conclusions

In this work, we successfully obtained a relatively flat paraffin coating with closed-type precise temperature-controlled evaporation and explored the adhesion effect on the hyperfine frequency shift of alkali metal vapor cell using the peak force tapping technique from simultaneous topography and adhesion information, which involves the energy transfer mechanism of the alkali atom colliding with vapor cell wall. We found the uniform and appropriate temperature of the surface benefit breaking the surface diffusion barrier, allowing the nucleated paraffin particles to aggregate and form a continuous paraffin coating. We also observed the different height of paraffin with the same adhesion and peak adhesion appear at the junction of paraffin particles and the substrate, which shows that the adhesion is only affected by the uniformity of the coating surface.

In addition, we investigated the energy transfer between the alkali atom and the coating by analyzing the force curve. We found that the surface adsorption energy of the coating can be calculated accurately with peak force tapping technology, moreover, it provides us a new method to test the quality of alkali atom vapor cell. While it is still necessary to further verify the energy with that from a quantum detection experiment, which is subject to subsequent research.

**Author Contributions:** Conceptualization, J.T. and Z.M.; Methodology, H.G. and Z.M.; Software, H.G.; Validation, J.T. and Y.L.; Formal analysis, J.L. and H.W.; Investigation, J.W.; Data curation, J.T.; Writing—original draft preparation, J.W.; Writing—review and editing, Z.M. and H.W.; Visualization, H.G.; Supervision, J.L., Y.L., and Y.S.; Project administration, Z.M. All authors have read and agreed to the published version of the manuscript.

**Funding:** The authors acknowledge the support from the Fund for National Key Research and Development Program of China (2017YFB0503100), Shanxi "1331 Project" Key Subjects Construction (1331KSC), the National Key Research and Development Program of China (2018YFF01012502), the National Natural Science Foundation of China (NSFC) (51727808, 61874100, 61503346, and 51635011), and the Natural Science Foundation of Shanxi (SXNSF) (201701D121080 and 201803D421037).

**Acknowledgments:** The authors would like to express their sincere thanks to LetPub (www.letpub.com) for its linguistic assistance during the preparation of this manuscript.

**Conflicts of Interest:** The authors declare no conflict of interest.

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
