# Peer review of "Adhesion Effect on the Hyperfine Frequency Shift of an Alkali Metal Vapor Cell with Paraffin Coating Using Peak-Force Tapping AFM"

_coatings, doi:10.3390/coatings10010084_

Round 1
Reviewer 1 Report
Maybe, it can be interesting to determine the adhesion of paraffin coating on Si substrates; to discuss the interfaces between these layers and for comparing the three coatings surface, using different methods such as contact angle measurements, FTIR analysis etc. ..
Author Response
Dear editor and referee
Thank you very much for your comments and advices.
We have revised the manuscript, according to the comments and suggestions of referee. Point by point replies to the referee comments were listed below. We have sent the final version and an additional copy of our manuscript with the changes highlighted.
Best wishes
Zongmin Ma
Response to Reviewer 1
Question 1: Maybe, it can be interesting to determine the adhesion of paraffin coating on Si substrates; to discuss the interfaces between these layers and for comparing the three coatings surface, using different methods such as contact angle measurements, FTIR analysis, etc. ..
Answer 1:
Thank you for your comment. It is good advice to investigate the paraffin coating of the alkali-metal vapor cell with different methods.
As discussed in [20] and [22], they investigated the coating of the alkali-metal vapor cell with contact angle measurements, FTIR (Fourier transform infrared spectroscopy), DSC (Differential scanning calorimetry), NEXAFS (Near edge x-ray absorption fine structure), XPS (X-ray photoelectron spectroscopy), etc.
However, it is much more important to reveal the adhesion effect of the paraffin coating of the alkali-metal vapor cell by peak-force tapping AFM technology. Because the energy loss caused by the adhesion effect can be evaluated, which can decrease the spin relaxation time and reduce the sensitivity of the alkali-metal vapor cell, which is discussed in the manuscript.

Reviewer 2 Report
This manuscript describes the adhesion effect on the hyperfine frequency shift of an alkali metal vapor cell with paraffin coating using peak-force tapping AFM. This manuscript was well organized and contained various experimental data obtained by analytical tools. I recommend that this manuscript can be published after minor revision.
1. In Page 7, line 224, in "In Fig. 6(g) and Fig. 6(f)", Fig. 6(g) must be changed to Fig. 6(e).
2. In Page 4, line 140, Please explain the reason why there is no particle and just crack-like trace in Fig. 4(c) unlike Fig. 4a, 4b.
3. I am just wondering which method among three methods for paraffin coating is much better for adhesion and surface morphology.
Author Response
Dear editor and referee
Thank you very much for your comments and advices.
We have revised the manuscript, according to the comments and suggestions of referee. Point by point replies to the referee comments were listed below. We have sent the final version and an additional copy of our manuscript with the changes highlighted.
Best wishes
Zongmin Ma
Response to Reviewer 2
Question 1: In Page 7, line 224, in "In Fig. 6(g) and Fig. 6(f)", Fig. 6(g) must be changed to Fig. 6(e).
Answer 1:
Thank you for your comment. We have changed “Fig. 6(g)” into “Fig. 6(e)”.
Question 2: In Page 4, line 140, Please explain the reason why there is no particle and just crack-like trace in Fig. 4(c) unlike Fig. 4a, 4b.
Answer 2:
Thank you for your comment.
It can be seen from Fig. 4(g) that the height difference of crack like trace in Fig. 4(c) is 0.8 nm, while the curvature radius of the probe tip is about 10 nm. Therefore, in the measurement of the force spectrum, the effect of surface fluctuation with 0.8 nm on the adhesion is less than 0.5 pN. So there is no obvious crack like trace in the analysis of adhesion measured from the force spectrum.
Question 3: I am just wondering which method among three methods for paraffin coating is much better for adhesion and surface morphology.
Answer 3:
As shown in Fig. 4, the adhesion and surface morphology of coatings produced by open-type evaporation, closed-type rapid cooling evaporation, and closed-type precise temperature-controlled evaporation are 70pN/192nm, 37pN/12nm, and 6pN/0.8nm, respectively. This means that the closed-type precise temperature-controlled evaporation is better than that compared to the other methods.

Reviewer 3 Report
"235 Paraffin coating can reduce the adhesion of the inner surface of the alkali-metal vapor cell, but
236 why it can extending spin relaxation time and increase the number of collisions, which involves the
237 transfer of energy during a collision." This sentence requires corrections to improve clarity.
Author Response
Dear editor and referee
Thank you very much for your comments and advices.
We have revised the manuscript, according to the comments and suggestions of referee. Point by point replies to the referee comments were listed below. We have sent the final version and an additional copy of our manuscript with the changes highlighted.
Best wishes
Zongmin Ma
Response to Reviewer 3
Question 1: "Paraffin coating can reduce the adhesion of the inner surface of the alkali-metal vapor cell, but why it can extending spin relaxation time and increase the number of collisions, which involves the transfer of energy during a collision." This sentence requires corrections to improve clarity.
Answer 1:
Thank you for your comment. We have corrected the sentence and rewritten it as follows:
“Paraffin coating can reduce the adhesion of the inner surface of the alkali-metal vapor cell, which can extend spin relaxation time and increase the number of collisions by reducing the energy loss during a collision.”
